# Post-Pandemic Mental Health: Psychological Distress and Burnout Syndrome in Regular Basic Education Teachers

Edwin Gustavo Estrada-Araoz [1,*], Judith Annie Bautista Quispe [2], Benjamin Velazco Reyes [3], Humberto Mamani Coaquira [4], Papa Pio Ascona Garcia [5] and Yessenia Luz Arias Palomino [6]

1   Facultad de Educación, Universidad Nacional Amazónica de Madre de Dios, Puerto Maldonado 17001, Peru
2   Facultad de Educación, Universidad Nacional del Altiplano, Puno 21001, Peru; jbautistaq@unap.edu.pe
3   Escuela Profesional de Arte, Universidad Nacional del Altiplano, Puno 21001, Peru; bvelazco@unap.edu.pe
4   Escuela Profesional de Educación Primaria, Universidad Nacional del Altiplano, Puno 21001, Peru; hmamanic@unap.edu.pe
5   Escuela Profesional de Ingeniería Civil, Universidad Nacional Intercultural Fabiola Salazar Leguía de Bagua, Bagua 01721, Peru; pascona@unibagua.edu.pe
6   Escuela Profesional de Educación Secundaria, Universidad Nacional del Altiplano, Puno 21001, Peru; yessenialuzarias@gmail.com
*   Correspondence: gestrada@unamad.edu.pe

**Abstract:** The COVID-19 pandemic has affected the mental health of regular basic education teachers. Despite the fact that in recent months the number of infections has decreased significantly, the return to face-to-face classes is of great concern to teachers due to the adverse educational context they must face. In this sense, the objective of this study was to determine whether or not psychological distress is significantly related to burnout syndrome in regular basic education teachers upon their return to face-to-face classes. This research employed a quantitative approach, the design was non-experimental, and the type of study was descriptive–correlational and cross-sectional. In total, 184 teachers participated and responded using the Psychological Distress Scale and the Maslach Burnout Inventory, which are instruments with adequate psychometric properties. The results indicated that 40.7% of the teachers had a moderate level of psychological distress, and 45.1% also had a moderate level of burnout syndrome. Likewise, it was found that the Pearson's r correlation coefficient between the variables psychological distress and burnout syndrome was 0.752, and the *p*-value was below the level of significance ($p < 0.05$). It was concluded that there is a direct and significant relationship between psychological distress and burnout syndrome in regular basic education teachers upon their return to face-to-face classes. For this reason, it is suggested that the Ministry of Education should design policies that allow a reassessment of the work that teachers have been carrying out and that promote the implementation of preventive and corrective programs to improve their mental health.

**Keywords:** psychological distress; burnout syndrome; mental health; teachers; post-pandemic

## 1. Introduction

The COVID-19 pandemic has caused many changes in people's lives (Osman et al. 2022). The worldwide spread of the virus starting in November 2019 increased the levels of concern in society, a situation that is understandable, since people were worried about their health and did not want to contract the virus, since the sequelae that the virus produces in the respiratory system increases the risk of death (Lin et al. 2020).

In the educational field, mandatory reforms were implemented for the purpose of avoiding crowds, reducing the spread of the virus, and providing continuity to the educational service that saw the migration from face-to-face education to the virtual modality. However, due to connectivity and accessibility problems on the part of the educational community, these changes did not have the expected results (Huanca-Arohuanca et al. 2020; Estrada et al. 2022b).

In Peru, in February 2022, the rate of infections and deaths caused by COVID-19 decreased significantly due to vaccination campaigns (Taborda et al. 2022). In this sense, many activities that were carried out virtually during the pandemic were carried out again in person. However, biosafety protocols were established for this. Likewise, in view of a favorable context, the Ministry of Education (MINEDU) established that starting in April 2022, classes would also return to face-to-face classes.

Since then, regular basic education teachers (professors who teach students in initial, primary, and secondary education) have been providing educational services to students who, after two years of virtual training, returned to face-to-face classes. However, during the virtual classes, many of the students did not develop the competencies, abilities, and skills established in the curriculum and that corresponded to the grades they were given due to the problems of connectivity and accessibility, the lack of support from their parents, and the poor methodology of some teachers when developing their learning sessions virtually.

However, it is evident that teachers are making more effort and are providing more attention and support to students who did not learn at the expected level during virtual classes. Likewise, there is pressure from parents for students to improve and develop skills and abilities according to their corresponding ages. In the same way, many activities and learning strategies have been re-evaluated to make them more relevant to students' characteristics and particularities (Villarreal 2023). This worrying, discouraging, and frustrating context has caused an increase in teachers' workloads, since they have to carry out additional work outside of their working days. Likewise, the erosion of their emotional resources has increased, and their mental health has been affected, causing, among other emotional problems, psychological distress and burnout syndrome.

Psychological distress is considered to be, according to the fifth edition of the Diagnostic and Statistical Manual of Mental Disorders (DSM-5), an undifferentiated group of symptoms ranging from symptoms of anxiety and depression to functional deterioration, traits of personality (confusing and worrisome) and behavior problems (American Psychiatric Association 2013). It is also associated with a state of emotional suffering characterized by symptoms of depression (loss of interest, unhappiness, or despair) and anxiety (restlessness or a feeling of tension). Additionally, it is characterized by other somatic symptoms such as insomnia, headaches, and a lack of energy that are likely to vary across different areas (Belay et al. 2021).

There are several variables that increase the possibility of suffering from psychological distress, from biological, psychological, socioeconomic, and work factors to lifestyle factors (Carod 2017). Psychological distress in Peruvian teachers of regular basic education is a common problem due to the high demand for cognitive, psychological, and affective resources to provide adequate attention to students with various characteristics and needs, which adds to their poor working conditions and is not sufficiently remunerated (Estrada et al. 2021).

Burnout syndrome was initially described by Freudenberg (1974) and was addressed in depth by Maslach (1982). It refers to a form of fatigue or exhaustion caused especially by the work carried out by professionals, such as teachers, doctors, nurses, and psychologists, among others, whose responsibilities include interacting with other people. In recent decades, interest in investigating this variable has grown due to the repercussions it has on the physical and psychological health of people, as well as their family, work, and social environments (Almeida et al. 2015).

Burnout is currently considered a chronic adjustment disorder (Chavarría et al. 2017) and tends to occur when people are exposed to various stressors for a long time, whether emotional or interpersonal, and a high emotional demand from people to whom they provide some kind of service (Domínguez et al. 2017; Magalhães et al. 2015).

Maslach and Jackson (1986) determined that burnout syndrome is made up of three components: emotional exhaustion, depersonalization, and personal accomplishment. Emotional exhaustion refers to the sustained depletion of energy or emotional resources that cause fatigue in a person. Depersonalization is associated with the development of negative

and indifferent feelings and attitudes towards other people. Personal accomplishment has to do with the cognitive self-assessment that professionals make about themselves and the work they carry out in their workplace.

However, teachers who suffer from this disease tend to provide a poor service to students, act indifferently, and do not offer support in the tasks assigned to them. In fact, its prevalence affects their performance in cases where the pathology of burnout is not identified and treated in a timely manner (Fuster et al. 2019).

Ourcilleón et al. (2007) considered that the presence of burnout in professionals has four main causes, which are divided into the following areas:

- Individual causes have to do with the feelings that teachers develop when they become too involved in the problems presented by students, which generates feelings of guilt, a subsequent decrease in levels of personal achievement, and an increase in fatigue.
- Emotional causes are related to the poor interpersonal relationships that teachers have with their colleagues, managers, and even their students, as well as the lack of support they receive from their immediate bosses (principals and deputy principals).
- Social causes are associated with the new policies and responsibilities that increase the bureaucratic burden of teachers, as well as demand, curricular changes, and new work methodologies.
- Organizational causes arise from problems within the organization of the educational institution, such as the lack of coordination and articulation between the different levels (early childhood education, elementary, and high school) as well as the poor exercise of leadership of the management team.

Some research has been carried out to determine whether or not psychological distress is related to burnout syndrome. Based on work in Nigeria, Ozoemena et al. (2021) concluded that there was a direct and significant relationship between both variables ($r = 0.330$); therefore, it was necessary to include in teacher training studies issues associated with the development of interpersonal skills, stress management skills, and resilience in order to prepare teachers for their future work. Similarly, in Malaysia, it was also reported that there was a direct relationship between both study variables (Hashim and Al 2022).

This research is relevant and original in the post-pandemic context, since it will allow the Ministry of Education and its decentralized offices to implement national, regional, and local policies that will lead to improvements in the working conditions in which teachers operate. On the other hand, the management teams of educational institutions will be able to develop preventive and corrective programs to promote the mental health of teachers.

Consequently, the objective was to determine whether or not psychological distress is significantly related to burnout syndrome in regular basic education teachers upon their return to face-to-face classes.

## 2. Materials and Methods

The research approach was quantitative, the design was non-experimental, and the type of study was descriptive–correlational and cross-sectional (Hernández and Mendoza 2018). The population consisted of 351 regular basic education teachers from the city of Puerto Maldonado (Peru). This Amazonian city is on the border with the countries of Brazil and Bolivia. Hence, it is a city with great demographic growth, and the school population is heterogeneous; that is, the students come from different Peruvian regions. The sample was made up of 184 teachers, a number determined by probabilistic sampling with a confidence level of 95% and a significance level of 5%. As seen in Table 1, 55.4% of the participants were women, and 44.6% were men. Regarding the age group, 51.1% were between 21 and 40 years old, while 48.9% were between 41 and 64 years old. Regarding the level of the educational system, 39.1% were elementary school teachers, 31% were early childhood education teachers, and 29.9% were high school teachers.

**Table 1.** Sociodemographic characteristics of the sample.

| Variables | Sociodemographic Characteristics | *n* = 184 | % |
|---|---|---|---|
| Gender | Male | 82 | 44.6 |
| | Female | 102 | 55.4 |
| Age group | From 21 to 40 years old | 94 | 51.1 |
| | From 41 to 64 years old | 90 | 48.9 |
| level of educational system | Early childhood education | 57 | 31.0 |
| | Elementary | 72 | 39.1 |
| | High school | 55 | 29.9 |

Surveys were used to collect data; to assess psychological distress and burnout syndrome, the Psychological Distress Scale and the Maslach Burnout Inventory were used, respectively. These instruments are used in the Peruvian and international context in various investigations to evaluate the previously mentioned variables.

The Psychological Distress Scale was designed by Kessler et al. (2003) and adapted to the Peruvian context by Arias et al. (2019). It assesses anxiety and depression through the frequency of non-specific symptoms during the last 30 days. It consists of 10 items that are quantitatively rated using a 3-point Likert scale ranging from 1 (never) to 3 (always). Its psychometric properties were determined through the validity and reliability process. In this sense, it was found that the aforementioned scale had an adequate level of validity based on the content (Aiken's V = 0.845) and reliability ($\alpha$ = 0.862).

The Maslach Burnout Inventory was prepared by Maslach and Jackson (1986) and evaluates the prevalence of burnout caused by work activities that workers usually develop. It is made up of 22 Likert-type items (never, sometimes times, and always) and measures 3 dimensions: emotional exhaustion (items 1 to 9), depersonalization (items 10 to 14), and personal accomplishment (items 15 to 22). Its psychometric properties were determined in a previous investigation carried out in Peru (Estrada and Gallegos 2020), where it was found that the inventory had an adequate level of validity based on the content (Aiken's V = 0.801) and reliability ($\alpha$ = 0.823).

The data collection process was carried out from July to August 2022, dates on which all the Peruvian educational institutions of regular basic education were providing educational service in person. For this purpose, authorization was requested from the Tambopata Local Educational Management Unit. Once authorization was obtained, the permission of the management staff of each educational institution was obtained, and a coordination meeting with the teachers was requested to define the days on which to conduct data collection. In this regard, informed consent was acquired from each participant, guaranteeing confidentiality and anonymity.

Data analysis was performed at descriptive and inferential levels. The descriptive analysis was developed through results elaborated using SPSS V.25 software. The inferential results were obtained using Pearson's r correlation coefficient. This statistic was relevant to determining whether or not the variables and dimensions were significantly related. Additionally, Student's T (t) parametric test was used to determine whether or not there were statistically significant differences regarding psychological distress and burnout syndrome according to gender. Finally, a simple linear regression analysis was performed in order to determine whether or not psychological distress indicated burnout syndrome.

Regarding ethical considerations, this research was carried out in accordance with the Declaration of Helsinki and was approved by the UNAMAD Ethics Committee, protocol code 002, in July 2022. Likewise, the teachers were informed about the purpose and nature of the research and provided their informed consent, which guaranteed the anonymous and voluntary nature of their participation.

## 3. Results

According to Figure 1, the level of psychological distress of 40.7% of the teachers was moderate, 35.3% was low, and 24% was high. These results indicate that the teachers showed feelings of sadness, uncertainty, confusion, worry, and hopelessness upon returning to face-to-face classes. On the other hand, it was found that 45.1% of the teachers presented a moderate level of burnout syndrome, 38.6% had a high level, and 16.3% had a low level, which indicated that teachers showed significant decreases in their energy and emotional resources, caused mainly by the presence of various stressors in the educational context. In the emotional exhaustion, depersonalization, and personal accomplishment dimensions, the moderate level predominated.

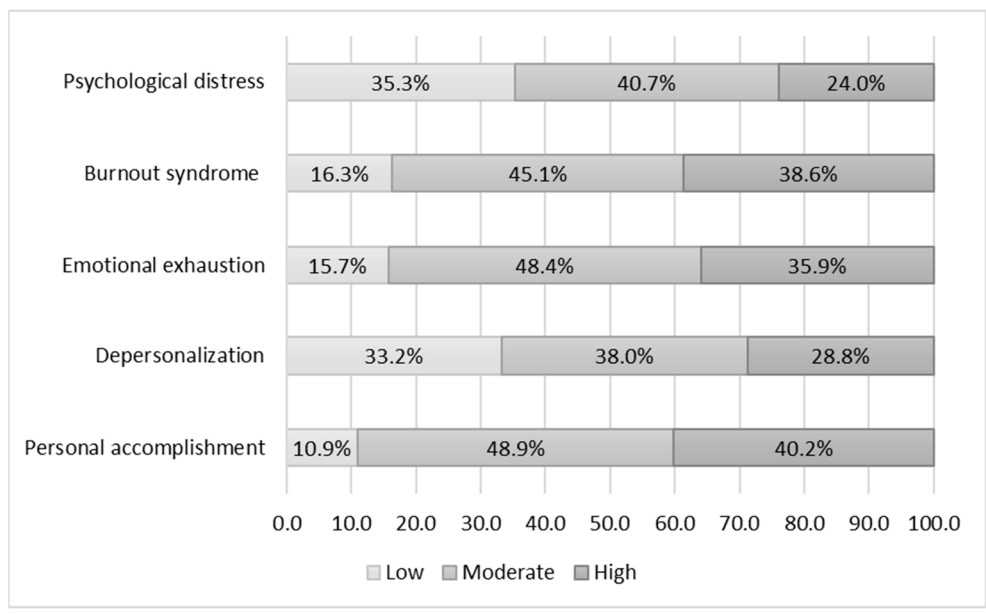

**Figure 1.** Descriptive results of the variables psychological distress and burnout syndrome.

Table 2 shows that the main symptoms associated with psychological distress expressed by teachers were feeling depressed, feeling that everything they did required a great effort, being desperate, feeling restless, and being sad.

**Table 2.** Answers to questions related to the Psychological Distress Scale.

| Items | Never | | Sometimes | | Always | |
|---|---|---|---|---|---|---|
| | *n* | % | *n* | % | *n* | % |
| Roughly, how often did you feel tired out for no good reason? | 88 | 47.8 | 71 | 38.6 | 25 | 13.6 |
| Roughly, how often did you feel nervous? | 56 | 30.4 | 90 | 48.9 | 38 | 20.7 |
| Roughly, how often did you feel so nervous that nothing could calm you down? | 85 | 46.2 | 70 | 38.0 | 29 | 15.8 |
| Roughly, how often did you feel hopeless? | 54 | 29.3 | 73 | 39.7 | 57 | 31.0 |
| Roughly, how often did you feel restless or fidgety? | 66 | 35.9 | 70 | 38.0 | 48 | 26.1 |
| Roughly, how often did you feel so restless you could not sit still? | 64 | 34.8 | 81 | 44.0 | 39 | 21.2 |
| Roughly, how often did you feel depressed? | 38 | 20.7 | 81 | 44.0 | 65 | 35.3 |
| Roughly, how often did you feel that everything was an effort? | 45 | 24.5 | 80 | 43.5 | 59 | 32.1 |
| Roughly, how often did you feel so sad that nothing could cheer you up? | 50 | 27.2 | 86 | 46.7 | 48 | 26.1 |
| Roughly, how often did you feel did you feel worthless? | 104 | 56.5 | 46 | 25.0 | 34 | 18.5 |

According to Table 3, the indicators associated with burnout syndrome most frequently reported by teachers were feeling exhausted at the end of their workday, being emotionally exhausted due to their work, and stress from working in direct contact with students.

**Table 3.** Answers to the questions related to the Maslach Burnout Inventory.

| Items | Never | | Sometimes | | Always | |
|---|---|---|---|---|---|---|
| | *n* | % | *n* | % | *n* | % |
| I feel emotionally exhausted because of my work. | 22 | 12.0 | 82 | 44.6 | 80 | 43.5 |
| I feel worn out at the end of a working day. | 20 | 10.9 | 80 | 43.5 | 84 | 45.7 |
| I feel tired as soon as I get up in the morning and see a new working day stretched out in front of me. | 36 | 19.6 | 96 | 52.2 | 52 | 28.3 |
| Working with students the whole day is stressful for me. | 35 | 19.0 | 100 | 54.3 | 49 | 26.6 |
| I feel burned out because of my work. | 31 | 16.8 | 83 | 45.1 | 70 | 38.0 |
| I feel frustrated by my work. | 34 | 18.5 | 90 | 48.9 | 60 | 32.6 |
| I get the feeling that I work too hard. | 21 | 11.4 | 103 | 56.0 | 60 | 32.6 |
| Being in direct contact with students at work is too stressful. | 25 | 13.6 | 85 | 46.2 | 74 | 40.2 |
| At work, I feel that I have reached the limit of my possibilities. | 36 | 19.6 | 82 | 44.6 | 66 | 35.9 |
| I think I treat some students with indifference. | 69 | 37.5 | 64 | 34.8 | 51 | 27.7 |
| I think I've been more insensitive to people since I've been doing this job. | 65 | 35.3 | 70 | 38.0 | 49 | 26.6 |
| I worry that this job is hardening me emotionally. | 51 | 27.7 | 71 | 38.6 | 62 | 33.7 |
| I really don't care what happens to some of the students I'm in charge of at the educational institution. | 63 | 34.2 | 85 | 46.2 | 36 | 19.6 |
| I feel like students blame me for some of their problems. | 57 | 31.0 | 60 | 32.6 | 67 | 36.4 |
| I can easily understand what my students think. | 15 | 8.2 | 96 | 52.2 | 73 | 39.7 |
| I deal very well with the problems that my students present to me. | 25 | 13.6 | 92 | 50.0 | 67 | 36.4 |
| I feel that I influence other people positively through my work. | 20 | 10.9 | 75 | 40.8 | 89 | 48.4 |
| I find myself having a lot of vitality. | 50 | 27.2 | 80 | 43.5 | 54 | 29.3 |
| I have the ability to create a relaxed atmosphere for my students. | 16 | 8.7 | 93 | 50.5 | 75 | 40.8 |
| I feel encouraged after working together with my students. | 14 | 7.6 | 89 | 48.4 | 81 | 44.0 |
| I have done many worthwhile things in this job. | 10 | 5.4 | 95 | 51.6 | 79 | 42.9 |
| I feel that I know how to adequately deal with emotional problems at work. | 11 | 6.0 | 100 | 54.3 | 73 | 39.7 |

Regarding the comparison of means, it was found that there were statistically significant differences for the variables psychological distress (t = −0.184; $p < 0.05$) and burnout syndrome (t = −0.773; $p < 0.05$) between men and women (Table 4). In this sense, we can see that women presented higher scores than men did in both variables.

**Table 4.** Differences between men and women regarding the study variables.

| Variables | Male | | Female | | T | *p* |
|---|---|---|---|---|---|---|
| | M | SD | M | SD | | |
| Psychological distress | 30.35 | 5.681 | 36.27 | 5.703 | −0.184 | 0.010 * |
| Burnout syndrome | 67.53 | 9.530 | 74.32 | 10.350 | −0.773 | 0.002 * |

* Statistically significant differences.

Data obtained through the Kolmogorov–Smirnov normality test showed that the magnitude of the test error for the psychological distress and burnout syndrome variables was higher than the significance level ($p > 0.05$), which indicated that the scores conformed to a normal distribution. Therefore, it was decided to use Pearson's r parametric test to determine whether or not the variables and dimensions were significantly related.

As seen in Table 5, it was determined that psychological distress was directly and significantly correlated with burnout syndrome (r = 0.752; $p < 0.05$), emotional exhaustion (r = 0.668; $p < 0.05$), and depersonalization (r = 0.701; $p < 0.05$). On the other hand, it was found that the correlation between psychological distress and personal accomplishment was inverse and significant (r = −0.639; $p < 0.05$). One aspect that draws attention is that the level of correlation between psychological distress and the dimensions of the burnout syndrome was high—almost comparable to the levels of correlation between the burnout syndrome and its own dimensions.

**Table 5.** Correlation matrix between the variables and dimensions of the study.

| Variables | 1 | 2 | 3 | 4 | 5 |
|---|---|---|---|---|---|
| Psychological distress | 1 | - | - | - | - |
| Burnout syndrome | 0.752 ** | 1 | - | - | - |
| Emotional exhaustion | 0.668 ** | 0.737 ** | 1 | - | - |
| Depersonalization | 0.701 ** | 0.766 ** | 0.740 ** | 1 | - |
| Personal accomplishment | −0.639 ** | −0.689 ** | −0.781 ** | −0.707 ** | 1 |

** The correlation is significant at the 0.01 level (bilateral).

Regarding the linear regression analysis, Table 6 shows that there was an adequate fit to the model (Test F = 21.380; $p < 0.05$); that is, psychological distress ($\beta = 0.748$; $p < 0.05$) significantly indicated teacher burnout syndrome (adjusted $R^2 = 0.560$). In addition, the t values of the beta regression coefficients of the predictor variable were significant ($p < 0.05$).

**Table 6.** Linear regression analysis between psychological distress and burnout syndrome.

| Predictors | B | Standard Error | β | T | *p* |
|---|---|---|---|---|---|
| (Constant) | 1.044 | 0.375 | | 2.542 | 0.000 |
| Psychological distress | 0.540 | 0.061 | 0.748 | 12.287 | 0.000 |

Dependent variable: burnout syndrome. Note: adjusted $R^2 = 0.560$; $p < 0.05$; F = 21.380.

## 4. Discussion

The closure of educational institutions and the subsequent migration to virtual education due to the COVID-19 pandemic affected students, since many of them did not have the same opportunities, tools, and access necessary to continue learning during the pandemic. In this sense, the work carried out by teachers when returning to face-to-face classes became more complex, since the academic performance of the students was not consistent with what was established in the curricula and the curricular planning documents from which they were derived. In this sense, the learning achieved through virtual classes was insufficient for many students. This situation also affected the mental health of teachers. Therefore, in the present investigation, we sought to determine whether or not psychological distress was significantly related to burnout syndrome in regular basic education teachers upon their return to face-to-face classes.

It was found that teachers showed moderate levels of psychological distress. The symptoms associated with psychological distress expressed by teachers were feeling depressed, feeling that everything they did required a great effort, being desperate, feeling restless, and being sad. In this regard, it is argued that this situation could have been exacerbated due to their concern about possible contagion, since when the educational institutions were opened, many did not comply with all the biosafety protocols (Estrada et al. 2022a).

The described findings coincide with those reported by Hutchison et al. (2022), who determined that teachers suffered from psychological distress upon returning to face-to-face classes. Similarly, the findings are related to research carried out in Japan by Wakui et al. (2021), where they analyzed the mental health of teachers who worked in person after the reopening of educational institutions and found that they suffered moderate levels of psychological distress. Similarly, in Mexico, an investigation was carried out to evaluate the mental health and psychological impact on teachers and students upon their return to face-to-face classes, and moderate levels of psychological distress, stress, and anxiety were reported due to possible COVID-19 infection and readaptation to this form of teaching (Armenta et al. 2023).

Psychological distress is an emotional state that has been used as an indicator of teachers' mental health. In addition, it has been argued that it is a momentary response to stress, and that if it is not treated promptly, it can become pathological, negatively affect mental health and personal well-being and increase the probability of developing physical illnesses in those who suffer from it (Deasy et al. 2014).

Another finding indicated that teachers suffered from moderate levels of burnout syndrome. In this sense, the teachers experience decreased energy and emotional resources, which were caused mainly by the presence of various stressors in the educational context, one of the main ones being the labor aspect. Currently, it is claimed that in the post-pandemic context, the workload of teachers had increased, since they employ additional school activities outside of their working day to make up for the learning that did not materialize during virtual classes (Villarreal 2023).

The results of this paper coincide with those reported by Villarreal (2023), who determined that the level of burnout in the sample of teachers ranged between mild and moderate levels. Among the main causes were the educational lags in students, and the pressure teachers felt to fill the academic gaps caused by virtual education. On the other hand, the level of burnout syndrome found was lower than that reported by Estrada and Gallegos (2020), who reported in their research that teachers were characterized by high levels of burnout syndrome (42.1%), emotional exhaustion (40.3%), and depersonalization (37.7%); however, they presented low levels of personal accomplishment (39.6%). These differences in the results may be explained by sociodemographic aspects and the academic conditions from which the data were obtained.

Teaching is considered a high-risk profession for suffering from burnout syndrome because it demands skills and commitments to carry out various activities inside and outside the institution, using time that should be devoted to rest and recreation by teachers (Rodríguez et al. 2017). In addition, the demands of the current context require teachers to train competent and comprehensive students. In this context, the pressure that exists on teachers to achieve greater learning and achieve educational objectives can harm their health, both physical and mental.

It was also found that women presented higher levels of psychological distress and burnout syndrome than men did, a finding that may have different reasons. There is some research indicating the externalization of emotional and physiological manifestations by women in stressful contexts (Vidal et al. 2018). Furthermore, in addition to their work responsibilities, they may assume additional tasks at home to a greater degree than men do, such as family responsibilities, childcare, and other domestic activities (Rodríguez et al. 2019). There is some research that also determined that women presented psychological distress and burnout syndrome with greater frequency and intensity compared to men (Ozoemena et al. 2021; Estrada and Gallegos 2020).

An important finding shows that psychological distress was directly and significantly related to burnout syndrome. In this sense, teachers who manifested high levels of psychological distress also showed high levels of burnout syndrome and, conversely, teachers who had low levels of psychological distress also presented low levels of burnout syndrome. There is some research that supports the above results. Conducting research in Nigeria, Ozoemena et al. (2021) concluded that there was a direct and significant relationship between both variables (r = 0.330). Therefore, it is necessary to include in teacher training studies issues associated with the development of interpersonal skills, stress management skills, and resilience to prepare them for the future work. Similarly, in Malaysia, it was also reported that there was a direct relationship between both variables (Hashim and Al 2022).

In Peru, the conditions in which teachers work are poor. According to a technical report prepared by the National Education Council, 60.5% of teachers did not have appropriate educational materials, which limited their pedagogical practice. Likewise, they dedicated approximately 12 additional hours to their working day during the week in order to prepare their classes and materials, as well as holding meetings with parents. As for their remuneration, 66.3% of them were dissatisfied with their remuneration, which was why they chose to look for alternative jobs to satisfy the basic needs of their families (Estrada and Gallegos 2021). These factors exacerbate the problems identified and increase the levels of dissatisfaction of teachers with the work they do.

This research addresses very important issues associated with the mental health of teachers. However, during the post-pandemic period and in the local, national, and

international contexts, they have been rarely studied, and as such, the findings are relevant and original. Despite this, it is necessary to mention that there are some limitations, which are detailed below. In the first place, the sample was homogeneous, since only teachers from the urban area and who had common sociodemographic characteristics participated. Secondly, the research design was cross-sectional, which did not allow causal relationships between variables to be established. Third, the data collection instrument was self-administered, a situation that could cause social desirability biases. These aspects do not allow generalizations to be made, and thus, caution is required when interpreting the results. Therefore, it would be important for future research to increase the sample size and include teachers from rural areas. Likewise, longitudinal designs should be used to establish causal and temporal relationships. Finally, it would be important to use alternative data collection techniques and instruments that complement and give greater objectivity to the aforementioned process.

## 5. Conclusions

Mental health is associated with a state of well-being in which each person is aware of their own potential, can cope with the normal stresses of life, can develop productively and fruitfully, and can be able to contribute to their own well-being and to society. In the present investigation, it was determined that, during their return to face-to-face classes, regular basic education teachers showed moderate levels of psychological discomfort and burnout syndrome. When comparing by gender, it was established that women presented slightly higher scores, both in psychological distress and in burnout syndrome, compared to men. On the other hand, it was determined that both variables were directly and significantly related ($r = 0.752$; $p < 0.05$). By virtue of the foregoing findings, it is necessary that the competent authorities (the Ministry of Education and its decentralized offices) provide care, prevention, and protection and approach the provision of services and strategies in a way that improves the health of teachers.

**Author Contributions:** Conceptualization, E.G.E.-A., J.A.B.Q. and B.V.R.; methodology, E.G.E.-A., H.M.C., P.P.A.G. and Y.L.A.P.; software, E.G.E.-A., J.A.B.Q., B.V.R. and H.M.C.; validation, E.G.E.-A. and J.A.B.Q.; for-mal analysis, P.P.A.G.; investigation, E.G.E.-A., J.A.B.Q., B.V.R., H.M.C. and Y.L.A.P.; resources, E.G.E.-A. and J.A.B.Q.; data curation, P.P.A.G. and Y.L.A.P.; writing—original draft preparation, E.G.E.-A., J.A.B.Q. and H.M.C.; writing—review and editing, E.G.E.-A., J.A.B.Q., B.V.R., H.M.C., P.P.A.G. and Y.L.A.P.; visualization, E.G.E.-A., J.A.B.Q., B.V.R. and H.M.C.; supervision, E.G.E.-A.; project administration, E.G.E.-A. All authors have read and agreed to the published version of the manuscript.

**Funding:** This research received no external funding.

**Institutional Review Board Statement:** The study was conducted in accordance with the Declaration of Helsinki, and approved by the Ethics Committee of UNAMAD with protocol code 002 and approved in July 2022.

**Informed Consent Statement:** Informed consent was obtained from all subjects involved in the study.

**Data Availability Statement:** Not applicable.

**Conflicts of Interest:** The authors declare no conflict of interest.

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
