# Peer review of "Post-Pandemic Mental Health: Psychological Distress and Burnout Syndrome in Regular Basic Education Teachers"

_socsci, doi:10.3390/socsci12050279_

Round 1

Reviewer 1 Report

See the attachment.

Author Response

Referee’ comments

Responses

Suggestion: Post-pandemic mental health: Psychological distress and burn-out syndrome in primary education teachers in Peru.

The suggestion is appreciated; however, the research does not only focus on primary school teachers, it also involved early childhood and secondary school teachers. For this reason, we consider that it would not be appropriate to modify the title.

Whereas authors do refer to the questionnaire items, they do not refer at all to the names of the questionnaires from which these items were isolated for the purposes of their study.

The observation described, as well as several observations, do not correspond to the present investigation.

The description of the sample’s demographic characteristics belongs in the Methodology chapter instead of under ‘Results’.

The observation described, as well as several observations, do not correspond to the present investigation.

As previously mentioned, several observations do not correspond to the revised manuscript, which addresses psychological distress and burnout syndrome.

Reviewer 2 Report

This is a lovely study aiming at a time-relevant topic that deserves policy and scientific attention. The background is well introduced. It describes the insufficient virtual learning experience during the pandemic due to opportunities, tools and access availability. It causes potential mental health challenges to the teachers which in turn negatively impacts the learning experience of students upon returning to face-to-face classes. The mental health status of teachers upon returning to face-to-face classes is also characterized in this study by the Psychological Distress Scale and the Maslach Burnout Inventory. The moderate levels psychological and burnout issues are clearly summarized and seem consistent with other studies. Also, between-gender disparities are also discovered with strong evidence. The authors also acknowledged the samping limitations and generalization challenges. The main argument is that "psychological distress was directly and significantly related to burnout syndrome", which is supported by study collected data. However, this is merely a demonstration of strong correlations between two mental health instruments in a certain population, which does not help to reach the conclusion that "the competent authorities provide care, prevention, protection and approach services and strategies to improve the health of teachers". Nor does the analysis fit the "returning to face-to-face classes" scenario. In order to support these statements, a study which compares teachers before/after returning to face-to-face classes, or a cohort study comparing schools with or without face-to-face classes arrangements is needed. In addition, a comparison of teachers' mental health state to the general public is needed to draw the conclusion. Statistics-wise, the study consists of solid and suitable methods, except for some suggestions as follows: 1. Table 2 normality tests for both instruments have quite low p-values. By definition, a survey instrument cannot have values below 0, and have an upper bound, so it is strongly recommended to include a nonparametric sensitivity analysis. Given the strong correlation, I would not expect a different result. 2. The correlation is actually very, very strong according to Table 4. The correlation between Psychological distress and the sub-scales of burnout syndrome is almost comparable with the correlation between burnout syndrome and its own sub-scales. We can point it out somewhere in the main text. 3. It is usually not necessary to include both Pearson correlation and a univariable linear regression. Peasrson correlation itself is a linear model, which will almost always be consistent with the results from a linear regression with only 1 dependent variable. 4. The term psychological distress "predicts" burnout syndrome may be too strong because "prediction" indicates chronological relationships. The linear model only reveals a linear relationship, rather than a prediction. Maybe "indicates" is a better word. This is merely my suggestion, not even a strong one.   Again, it is a nice study addressing an issue that needs attention in a timely manner. However, the correlation analyses only shows the relationship between the two mental health instruments, which does not directly support the authors' claims. The descriptive results actually may support the claims with additional discussions and comparisons to external data sources.

Author Response

Referee’ comments

Responses

Table 2 normality tests for both instruments have quite low p-values. By definition, a survey instrument cannot have values below 0, and have an upper bound, so it is strongly recommended to include a nonparametric sensitivity analysis. Given the strong correlation, I would not expect a different result.

At the request of reviewer 3, Table 2 was removed because he stated that it was not relevant to consider. It was only reported in the text of the manuscript.

The correlation is actually very, very strong according to Table 4. The correlation between Psychological distress and the sub-scales of burnout syndrome is almost comparable with the correlation between burnout syndrome and its own sub-scales. We can point it out somewhere in the main text.

We appreciate the correction. In this sense, said information was added to the results, before the correlation matrix (it is in red).

It is usually not necessary to include both Pearson correlation and a univariable linear regression. Pearson correlation itself is a linear model, which will almost always be consistent with the results from a linear regression with only 1 dependent variable.

We appreciate the reviewer's correction; however, we believe that the coefficient of determination allows us to know to what extent one variable explains the behavior of the other variable. Therefore, we believe that it is not necessary to remove the table and information regarding the aforementioned coefficient (R2).

The term psychological distress "predicts" burnout syndrome may be too strong because "prediction" indicates chronological relationships. The linear model only reveals a linear relationship, rather than a prediction. Maybe "indicates" is a better word. This is merely my suggestion, not even a strong one.

We appreciate the correction. Therefore, we correct the term and replace it (it is in red).

Reviewer 3 Report

This paper reports an analysis of psychological distress and burnout in teachers. While it is generally well-written, some claims were made which required supporting evidence. There was also the suggestion of directionality in the findings and discussion which cannot be made using cross-sectional data. I have made specific comments below by section:

Introduction

1. Paragraph 5 - please elaborate on how is it evident that teachers are making more effort

2. Please define the term "basic education teachers"

Materials and methods

3. Can you give more details on the city in which the research was conducted for international readers?

4. Please provide an ethics reference number

5. Please clarify how you determined criteria for psychological distress and burnout

Results

6. I am not sure that all of the tables are required (e.g. table on normality) as these can just be reported in the text

Discussion

7. The second paragraph reads as though face-to-face classes cause distress, however, this cannot be claimed because the data is cross-sectional

8. Could you explain further why the prevalence of burnout syndrome is lower compared to Estrada and Gallegos?

9. You could be more explicit about the limitations, including the limitations of cross-sectional research

Author Response

Referee’ comments

Responses

1. Paragraph 5 - please elaborate on how is it evident that teachers are making more effort

We thank the reviewer for his observation. It has been specified which actions lead to teachers making a greater effort to return to face-to-face (indicated in red).

2. Please define the term "basic education teachers".

We thank the reviewer for his observation. It has been briefly specified what "basic education teachers" refer to (it is in parentheses and in red).

3. Can you give more details on the city in which the research was conducted for international readers?

We thank the reviewer for his observation. Brief characteristics of the city where the investigation was carried out have been specified (indicated in red).

4. Please provide an ethics reference number

We thank the reviewer for his observation. The code of ethical reference has been specified (indicated in red).

5. Please clarify how you determined criteria for psychological distress and burnout.

We thank the reviewer for his observation. It has been specified how both variables were evaluated (indicated in red).

6. I am not sure that all of the tables are required (e.g. table on normality) as these can just be reported in the text

We thank the reviewer for his observation. Table 2 has been deleted, however, information on the normality test is required in the text.

7. The second paragraph reads as though face-to-face classes cause distress, however, this cannot be claimed because the data is cross-sectional

We thank the reviewer for his observation. In this sense, it was clarified that what was found was during the return to the main classes. Likewise, the symptoms most frequently suffered by teachers were modified based on the descriptive analysis of the items that was carried out.

8. Could you explain further why the prevalence of burnout syndrome is lower compared to Estrada and Gallegos?

We thank the reviewer for his observation. The reason for the differences between our findings and what was reported by Estrada & Gallegos (2020) was clarified (indicated in red).

9. You could be more explicit about the limitations, including the limitations of cross-sectional research

We thank the reviewer for his observation. Limitations were detailed, including the issue of cross-sectional research design (indicated in red).

Round 2

Reviewer 1 Report

I accept the changes of the revised manuscript. 

Author Response

Reviewer 1 made no further comments. We appreciate the comments made in the first round.

Reviewer 3 Report

Thank you for sending your revised manuscript, and addressing my comments. I have a couple of further comments regarding some of the statements made in the introduction and discussion which are outlined below:

1. In the introduction, in paragraph 5, more information has been added around how teachers have made more effort etc but has not provided references to support this. Please clarify.

2. In the discussion, in paragraph 2, this still reads as though face-to-face classes caused moderate levels of psychological distress when it is not possible to come to this conclusion due to the design of the research. Please amend the wording throughout the manuscript so that it is clear that causality was not assessed/determined.

Author Response

Puerto Maldonado (Peru), April 30, 2023

Mr. Social Sciences Journal

Good morning, I am contacting you to respond to the comments made by Reviewer 3 about the research article that we submitted to the Behavioral Sciences journal. Below we detail, point by point, the answers. We hope to clarify the doubts and observations provided by the reviewer. Thank you very much for your attention.

Reviewer 3

Referee’ comments

Responses

1. In the introduction, in paragraph 5, more information has been added around how teachers have made more effort etc. but has not provided references to support this. Please clarify.

Regarding the first point, a quote (Villarreal, 2023) was included to clarify who made such a statement. It was highlighted in yellow.

2. In the discussion, in paragraph 2, this still reads as though face-to-face classes caused moderate levels of psychological distress when it is not possible to come to this conclusion due to the design of the research. Please amend the wording throughout the manuscript so that it is clear that causality was not assessed/determined.

The part where the levels of psychological distress and burnout syndrome were linked to the return to face-to-face classes was eliminated. Instead, a comparison was made with previous research, which did find that causal relationship.  It was highlighted in yellow.

Sincerely
